# Quality of life and its related psychological problems during coronavirus pandemic

Atefeh Homayuni[1], Zahra Hosseini [iD][2]*, Nahid Shahabi[1], Roghayeh Ezati Rad[1], Farah Moayedi[3]

1 Student Research Committee, Hormozgan University of Medical Sciences, Bandar Abbas, Iran,
2 Associate Professor of Health Education and Promotion, Social Determinants in Health Promotion Research Center, Hormozgan Health Institute, Hormozgan University of Medical Sciences, Bandar Abbas, Iran, 3 Tobacco and Health Research Center, Hormozgan University of Medical Sciences, Bandar Abbas, Iran

* hosseinishirin@ymail.com

## Abstract

### Background

The prevalence of coronavirus disease 2019 (COVID-19) has endangered the psychological health of individuals. This study aimed to assess the quality of life and its related psychological problems during COVID-19 pandemic.

### Methods

In this cross-sectional study, 559 citizens above the age of 16 years, in Isfahan and Bandar Abbas cities in Iran were selected with a convenient sampling method. An online questionnaire was used to collect the data, which consisted of five sections: demographic information, short health anxiety inventory (SHAI), perceived stress scale (PSS), world health organization quality of life questionnaire (WHOQOL-BREF) and Padua inventory. Data were analyzed using statistical tests including t-test, path analysis and structural equation modeling (SEM) using SPSS 24 and Amos 21 statistical software.

### Results

A total of 559 subjects with the mean age of 37.34 ± 11.19 years participated in this study. Most of the participants were female (78.5%), married (71.6%) and employed (40.9%). The majority of them also had a bachelor's degree (42.9%). There were significant negative correlations between perceived helplessness ($r$ = -.597, p = .000), perceived stress ($r$ = -.715, p = .000), risk of disease ($r$ = -.302, p = .000), negative effect of disease ($r$ = -.424, p = .000), health anxiety ($r$ = -.366, p = .000), contamination obsessions ($r$ = -.187, p = .000) and washing compulsions ($r$ = -.193, p = .000) with quality of life. On other hand, significant positive correlation was found between perceived self-efficacy ($r$ = .665, p = .000) and quality of life.

### Conclusions

According to our findings, health anxiety, perceived stress and obsessive-compulsive disorder were negatively affected psychological health during COVID-19 which in turn decreased

**Data Availability Statement:** All relevant data are within the paper and its Supporting Information files.

**Funding:** The authors received no specific funding for this work.

**Competing interests:** The authors have declared that no competing interests exist.

quality of life. Therefore, we suggest considering prevention and treatment of theses psychological problems to diminish the risk of reduced quality of life during COVID-19 global pandemic crisis.

## Introduction

Coronavirus disease 2019 (COVID-19) pandemic was first seen in December 2019 in china. The virus spread rapidly around the world and it was recognized as a pandemic by the World Health Organization in March 2020 [1]. The most common symptoms of the COVID-19 include fever, cough, shortness of breath, and gastrointestinal symptoms in some cases. In more severe cases, it may cause Acute Respiratory Distress Syndrome (ARDS) and hypoxic respiratory failure, which is a major cause of death in COVID-19 patients [2].

The first case of this disease in Iran was diagnosed in February 19, 2020, when two cases of COVID-19 were reported in Qom, and then the disease spread throughout Iran [3]. Based on the World Health Organization, up to October 4, 2022, there have been 7550021 confirmed cases of COVID-19 with 144448 deaths in Iran [4].

COVID-19 has imposed a serious threat to physical health and life and has caused a wide range of psychological and physical problems. Some factors such as fear of being infected with the virus, fear of death, spreading false news and rumors, interference in daily activities, travel prohibitions or restrictions, reduced social relationships (colleagues, friends and family), job and financial problems, etc endangered the mental health of people [5]. Thus, studying the psychological effects of this viral disease on the mental health at different levels of society is crucial. Many studies that investigated the psychological disorders of people in quarantine, have reported mental issues such as emotional disorder, depression, stress, irritability, insomnia, decreased attention, post-traumatic stress disorder, anger and emotional blunting [6–8].

The COVID-19 pandemic has had great disruptive impacts on communities throughout the world due to fear and stress, among which are fears for unemployment due to COVID-19-related restrictions, financial worries, and health-related worries. These stresses and worries may reduce the quality of life [9].

The most common triggers of psychological disorders and stress in people quarantined due to COVID-19 disease have been reported to be fear of being infected or infecting others, long period of quarantine, inadequate support and lack of access to care adequate medicine and food, and fatigue and boredom due to quarantine and isolation [10]. Zarabadipour et al. showed that access to social response systems, considerable weight change and reluctance to perform daily activities were among the factors, which had a significant relationship with stress scores among medical staff during COVID-19 pandemic. In their study, gender, the presence of high-risk symptoms among elderly individuals in the family, having a history of a decreased person due to COVID-19 in the family, exacerbation of the chronic disease, job closure, etc. had significant relationships with stress score [11]. Qi et al. [12] in their study among Chinese adults during the COVID-19 pandemic reported that perceived stress scores were negatively correlated with physical component summary score ($r = -.4$) and mental component summary score for health related quality of life ($r = -.6$). It indicates that participants who reported higher health-related quality of life, experienced lower levels of perceived stress.

Health anxiety occurs when mild physical changes are interpreted as a sign of a disease (usually a serious and dangerous disease) [13]. Most people experience some degree of health anxiety during their lifetime. This awareness can be a protective factor against health threatening factors and play a major role in identifying the early symptoms of the health issue and

enhancing health-promoting behaviors [14]. However, its excessive level can be dangerous and harmful to the person [13]. Recent studies in COVID-19 pandemics have shown that the frequency and severity of health anxiety have increased significantly [15]. In the case of infectious diseases or pandemic conditions, people with high health anxiety are prone to misunderstanding and they consider all these changes as evidence for an infectious disease [16]. People with high health anxiety also engage in a variety of non-adaptive safety behaviors such as over-washing of hands, social isolation, and shopping accompanied by fear. Low levels of health anxiety can also have negative effects on health behavior. People who find themselves at low risk for disease are also less likely to change their social behavior and ignore recommendations for social distance. Thus, health anxiety is one of several psychological factors that affect one's way of reaction to the spread of COVID-19 [17]. The results of a cross-sectional study conducted by Abdelghani et al. [18] on health care workers in hospitals of Egypt showed that health anxiety to COVID-19 virus was inversely correlated with all domains of quality of life (physical health, $r$ = -.409; psychological health, $r$ = -.453; social relationships, $r$ = -.224 and environmental health, $r$ = -.385).

Disease-related obsessive-compulsive disorder is another common psychological disorder during the pandemic. Information provided by various sources about the methods of infection with the virus may increase people's fear of this disease infection. Fear is a defense mechanism that a person shows against dangerous situations and includes the basic reactions necessary to survive from these threatening situations. However, fear that is not appropriate to the current situation may lead to a variety of psychological disorders such as obsessive-compulsive disorder, anxiety, depression and panic [19]. According to the American Psychological Association, fear of contamination is the most common obsession. Fear of contamination is often associated with mandatory rituals such as hand washing, cleaning, and taking appropriate action to reduce exposure to contaminated sources [20]. Since observing personal hygiene (regular hand washing, use of masks and gloves) and maintaining social distance are among the most important ways to prevent the spread of COVID-19 virus, the risk of practical obsessions during COVID-19 pandemic are considered very likely [21]. 60.3% of participants in the Abba-Aji et al.'s study [22] reported onset of obsessive-compulsive disorder symptoms (obsessions related to contamination with dirt, germs or viruses) and 53.8% had compulsions to wash hands repeatedly or in a special way, during the COVID-19 pandemic. Respondents who showed obsessive-compulsive disorder symptoms only since the start of COVID-19 pandemic were significantly more likely to have moderate/high stress.

Delay in starting vaccination, people's negligence in observing health protocols and their resistance to vaccination led to higher prevalence and mortality due to the COVID-19 in Iran, and as a result, imposed extra psychological pressure on people and decreased their quality of life. Residents of Bandar Abbas (a city with commercial ports) and Isfahan (an industrial city and a city with a lot of tourism attractions) provinces are more vulnerable to COVID-19. Since, the first step in creating effective interventions to promote people's quality of life during COVID-19 is to identify the relevant and predictive psychological factors. Therefore, the study was developed to respond to this question: Are there relationships between perceived stress, health anxiety and obsessive-compulsive disorder with quality of life during COVID-19 pandemic?

The following hypotheses are proposed:

1. There is a significant negative relationship between perceived stress and quality of life during COVID-19 pandemic; 2. There is a significant negative relationship between health anxiety and quality of life during COVID-19 pandemic; 3. There is a significant negative relationship between obsessive-compulsive disorder and quality of life during COVID-19 pandemic.

## Materials and methods

### Study design and population

This descriptive-correlational study was conducted from October 22, 2020 to February 3, 2021. The statistical population of this study consisted of residents over 16 years old in Isfahan and Bandar Abbas (in Iran).

### Sample size and sampling procedure

The sample size was calculated based on the following formula:

$$n = [(z_{1-\alpha/2} + z_{1-\beta})/C]^2 + 3, \ \ C = 0.5^*\ln[(1+r)/(1-r)]$$

Assuming a 5% error, 90% test power and a correlation coefficient of -0.18 in previous studies [23], the sample size was estimated as 320 people based on the formula. Considering the design effect of 1.2 for cluster sampling, the final sample size was estimated as 384.

The inclusion criteria for the sample selection included: literacy, access to internet to answer questions, not suffering from psychological disorders such as obsessive-compulsive disorder, the minimum age of 16 years, living in Isfahan or Bandar Abbas and willingness to participate in the study. Exclusion criteria were people with psychological disorders such as obsessive-compulsive disorder and unwillingness to participate in the study.

Regarding to the existing limitations due to the outbreak of COVID-19 and the impossibility of distributing questionnaires in paper form, online questionnaires were designed to collect the data. Questionnaires were distributed to people from Bandar Abbas and Isfahan provinces through various social media platforms (WhatsApp, Telegram, LinkedIn), email, channels and news agencies, public relations of University of Medical Sciences, Red Crescent, Municipality and University Student Research Committee. On the first page of the questionnaire, the purpose of the study was clearly explained to the participants and they were reminded that mentioning their first and last names would not be necessary and their information would be kept confidential.

### Ethics statement

Ethical approval was received for this study from the Ethics Committee of the Hormozgan University of Medical Sciences (IR.HUMS.REC.1399.439). Written informed consent was obtained from individuals who participated in this study. For participants between 16–18 years of old, the research ethics committee waived the requirement for parental consent.

### Research tools

The following demographic questionnaire and self-report tools were used to measure variables and collect data. The assessed demographic characteristics were gender, marital status, education level, age, employment status, history of infection with COVID-19 and chronic diseases (such as hypertension, diabetes, cardiovascular disease, kidney disease and etc).

**Short Health Anxiety Inventory (SHAI).** This inventory is an 18-item self-report scale. Each of the 18 items consists of four statements, in which individuals select the one that best reflects their feelings during the last six months. Items assess worry about health, awareness of bodily sensations or changes, and feared consequences of having an illness. The responses are scored on a 4-point Likert scale (0–3). This questionnaire enables estimating the levels of health anxiety in two components: the probability of getting the disease (14 questions) and the negative consequences of getting the disease (4 questions). Overall scores ranged from 0 to 54,

whereby the higher scores indicated the higher health anxiety [24]. Several studies have shown that this scale has good internal consistency, as its internal consistency coefficients ranged from .71 to .95. In the same studies, the reliability coefficient through test-retest method with a one-week interval has been obtained 0.90 [25]. This inventory was translated and validated in Iran by Karimi et al. [26] and its internal consistency was obtained .79. In the present study, Cronbach's alpha coefficients for the probability of getting the disease, the negative consequences of getting the disease and health anxiety were .833, .7 and .855, respectively.

**Perceived Stress Scale (PSS).**  The Perceived Stress Scale is a 14-item self-report tool developed by Cohen et al. (1983) to measure general perceived stress, thoughts and feelings about stressful events, control, overcoming, coping with stress, and experienced stress over the past month. On this scale, individuals are asked to indicate their feelings over the past 4 weeks on a 5-point Likert scale from 0 (never) to 4 (always). For example, "In the past month, how often have you felt that you were unable to control the important things in your life?" This scale measures two subscales: a) Perceived helplessness, which includes items 1, 2, 3, 8, 11, 12, and 14. B) Perceived self-efficacy, which includes items 4, 5, 6, 7, 9, 10, and 13, and these items are scored inversely. The lowest score is zero and the highest score is 56. The higher scores indicated the higher perceived stress. In a study conducted by Cohen et al. (1983), the internal consistency coefficients for each of the subscales and the general PSS were between .84 to .86 [27]. The reliability coefficients of the internal consistency for this scale were obtained .84 to .86 through Cronbach's alpha coefficient in two groups of students and one group of smokers in the smoking cessation program. Perceived stress scale is significantly correlated with life events, depressive and physical symptoms, social anxiety, and low life satisfaction [27]. This scale was translated and validated in Iran by Maroufizadeh et al. [28] and its internal consistency was obtained .9. In the present study, Cronbach's alpha coefficients for perceived helplessness, perceived self-efficacy and perceived stress were .844, .859 and .887, respectively.

**World Health Organization Quality of Life Questionnaire (WHOQOL-BREF).**  The Persian version of the WHOQOL-BREF was used to collect information. This questionnaire is a 26-item instrument consisting of four domains: physical health (7 items), psychological health (6 items), social relationships (3 items) and environmental health (8 items). The first two questions assess the general health and quality of life. Each question is scored on a 5-point Likert scale (1: strongly disagree to 5: strongly agree). The score of each aspect is calculated separately from the total score of its questions, so that score 4 indicates the worst and score 20 indicates the best situation in that aspect [29]. In this study, the standardized version of this questionnaire (developed by Usefy et al. [30]) was used. Cronbach's alpha coefficient for physical health, psychological health, social relationships and environmental health were .81, .72, .78, and .76, respectively. In the present study, Cronbach's alpha coefficients for physical health, psychological health, social relationships, environmental health and quality of life were .799, .849, .768, .812 and .932, respectively.

**Padua inventory.**  To measure washing compulsions and contamination obsessions, Padua inventory (Modified by Washington State University) was used [31]. Subjects' responses to each item are scored on a 5-point Likert scale (0 = not at all, 1 = low, 2 = somewhat, 3 = high, and 4 = very high). High scores indicated a high level of obsessive-compulsive disorder in the subject. Van Open [32] obtained obsessive-compulsive disorder coefficients at .94 for Padua questionnaire and above .80 for its subscales. This inventory was translated and validated in Iran by Shams et al. [33] and its internal consistency was obtained .92. In the present study, Cronbach's alpha coefficients for contamination obsessions, washing compulsions and obsessive-compulsive disorder were .823, .771 and .875, respectively.

## Data analysis

Data were analyzed in SPSS-24 and Amos-21 statistical softwares. Demographic characteristics were described using descriptive statistics including frequency and percentage. Mean, standard deviation, minimum and maximum values were calculated for continuous variables. To test the hypotheses and discover the relationships between the variables, path analysis and structural equation modeling (SEM) were used. The level of significance was considered to be 95% ($p < 0.05$).

# Results

## Participants' characteristics

A total number of 559 questionnaires were completed and returned. The participants' demographic information is as follows: 21.5% of the participants were male and 78.5% were female. More than half of the participants were married (71.6%) and the rest were single (26.8%), divorced (1.1%) or widow (0.5%). The majority of them had bachelor's degree (42.9%). In relation to job situation, 40.9% of the participants were employees. Only 14.5% of the participants were members of the medical, health and treatment staff and 83.4% of them had no history of chronic diseases (Table 1).

## Descriptive statistics

Descriptive statistics (mean, standard deviation, minimum and maximum values) of research variables are reported in Table 2.

**Table 1. Participants' demographic information (N = 559).**

| Characteristics | Categories | n(%) |
|---|---|---|
| Gender | Male | 120 (21.5) |
| | Female | 439 (78.5) |
| Marital status | Single | 150 (26.8) |
| | Married | 400 (71.6) |
| | Widow | 3 (0.5) |
| | Divorced | 6 (1.1) |
| Level of education | High school | 36 (6.5) |
| | Diploma | 100 (17.9) |
| | Associate degree | 43 (7.7) |
| | Bachelor's degree | 240 (42.9) |
| | Master's degree and higher | 140 (25) |
| Job situation | Student | 20 (3.6) |
| | University student | 39 (7) |
| | Unemployed | 25 (4.5) |
| | Housewife | 181 (32.4) |
| | Employee | 229 (40.9) |
| | Self-employed | 65 (11.6) |
| Smoking | Only cigarette | 23 (4.1) |
| | Only hookah | 22 (3.9) |
| | Both of them | 5 (0.9) |
| | None of them | 509 (91.1) |
| Medical staff | Yes | 81 (14.5) |
| | No | 478 (85.5) |
| Chronic diseases | Yes | 93 (16.6) |
| | No | 466 (83.4) |

**Table 2. Descriptive statistics of research variables.**

|  | Minimum | Maximum | Mean | Std. Deviation |
|---|---|---|---|---|
| age | 16 | 69 | 37.34 | 11.19 |
| Quality of life | 37.00 | 128.00 | 87.03 | 15.81 |
| Physical health | 8.00 | 35.00 | 24.35 | 4.80 |
| Psychological health | 7.00 | 30.00 | 19.15 | 4.56 |
| Social relationships | 3.00 | 15.00 | 10.01 | 2.53 |
| Environmental health | 11.00 | 40.00 | 26.26 | 5.21 |
| Obsessive-compulsive disorder | .00 | 40.00 | 19.70 | 9.12 |
| Contamination obsessions | .00 | 24.00 | 10.69 | 5.54 |
| Washing compulsions | .00 | 16.00 | 9.01 | 4.53 |
| Health anxiety | .00 | 47.00 | 14.15 | 7.67 |
| Risk of disease | .00 | 40.00 | 12.32 | 6.42 |
| Negative effect of disease | .00 | 12.00 | 1.83 | 2.06 |
| Perceived stress | 2.00 | 54.00 | 27.4955 | 8.83518 |
| Perceived helplessness | .00 | 28.00 | 13.92 | 5.41 |
| Perceived self-efficacy | .00 | 28.00 | 14.43 | 4.65 |

As the results showed, the mean and standard deviation scores of perceived stress were (27.49±8.83), health anxiety (14.15±7.67), obsessive-compulsive disorder (19.7±9.12) and quality of life (87.03±15.81).

## Correlations

The correlation matrix of the variables is presented in Table 3.

Results of Pearson correlation showed that there were significant negative relationships between the independent variables (health anxiety, perceived stress, contamination obsessions and washing compulsions) with quality of life.

## Structural equation modeling results

The fit indices of the model are presented in Table 4.

According to the results of Table 4, the fit indices to evaluate the totality of the final model indicated that in general the model has appropriate fitness.

**Table 3. Correlation coefficients of study variables.**

| variables | 1 | 2 | 3 | 4 | 5 | 6 | 7 | 8 | 9 |
|---|---|---|---|---|---|---|---|---|---|
| Perceived helplessness |  | 0.541** | 0.897** | 0.327** | 0.303** | 0.355** | 0.147** | 0.19** | -0.597** |
| Perceived self-efficacy |  |  | -0.857** | -0.186** | -0.279** | -0.231** | -0.071 | -0.094* | 0.665** |
| Perceived stress |  |  |  | 0.298** | 0.332** | 0.338** | 0.127** | 0.166**** | -0.715** |
| Risk of disease |  |  |  |  | 0.508** | 0.973** | 0.328** | 0.281** | -0.302** |
| Negative effect of disease |  |  |  |  |  | 0.693** | 0.221** | 0.18** | -0.424** |
| Health anxiety |  |  |  |  |  |  | 0.334** | 0.283** | -0.366** |
| Contamination obsessions |  |  |  |  |  |  |  | 0.634** | -0.187** |
| Washing compulsions |  |  |  |  |  |  |  |  | -0.193** |
| Quality of life |  |  |  |  |  |  |  |  |  |

**Correlation is significant at the 0.01 level.

* Correlation is significant at the 0.05 level.

**Table 4. Model fit index of predictive pattern of quality of life.**

| RMSEA | TLI | GFI | NFI | IFI | CFI | CMIN/DF |
|---|---|---|---|---|---|---|
| 0.085 | 0.967 | 0.976 | 0.978 | 0.983 | 0.982 | 5.054 |

The results of structural equation modeling are presented at Fig 1.

First, the regression model is fitted. The regression coefficients given in Table 5 indicate the effectiveness or non-effectiveness of the subscales on the main variable. The results showed that the factor loading of obsessive-compulsive disorder is -0.068 (p = 0.049) and this variable should be removed from the model because of its low factor loading. The factor loading of health anxiety is -0.189 (p = 0.001) and this variable should also be removed from the model. But the factor loading of the perceived stress with quality of life is -0.98 (p < 0.001).

The structural equation model, after removing obsessive-compulsive disorder and health anxiety variables, is presented in Fig 2.

First, the regression model is fitted. The regression coefficients given in Table 6 indicate the effectiveness or non-effectiveness of the subscales on the main variable. The perceived stress has a significant level of *** (less than 0.001) with quality of life and its factor loading is -1.000. The subscales of environmental health with a standardized estimate of 0.744, social relationships with a standardized estimate of 0.756, psychological health with a standardized estimate of 0.927, and physical health with a standardized estimate of 0.762 explain the quality of life. Perceived helplessness with a standardized estimate of 0.668 and perceived self-efficacy with a standardized estimate of -0.717 explain the perceived stress.

## Discussion

The present study assessed the quality of life and its related psychological problems during COVID-19 pandemic. The results indicated that there were significant negative correlations between health anxiety, perceived stress and obsessive-compulsive disorder with quality of life.

### The correlation between health anxiety and quality of life

The results showed a significant negative correlation between health anxiety and quality of life. The current finding is consistent with findings reported by Abdelghani et al. [18], Hayter et al.

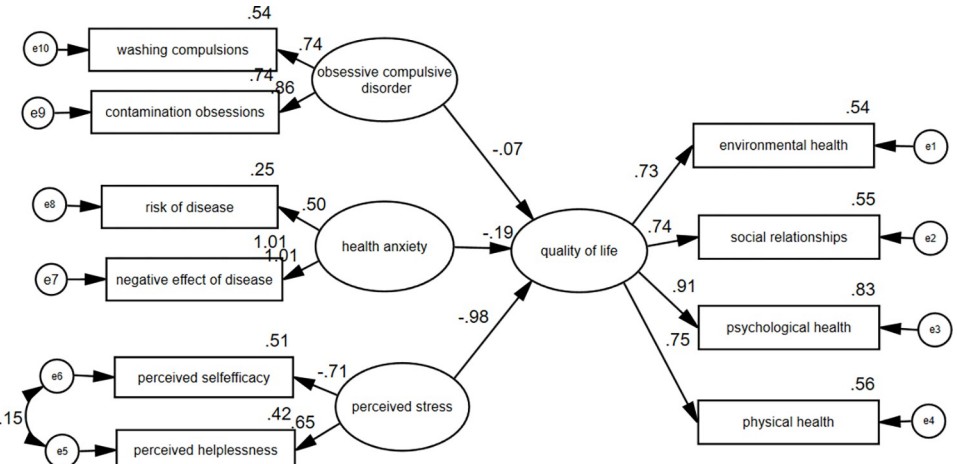

**Fig 1. The structural equation modeling (SEM) of the relationships between psychological variables and quality of life.**

**Table 5. Regression weight & standardized regression weight.**

| | | | Estimate | S.E. | C.R. | P | Standardized Estimate |
|---|---|---|---|---|---|---|---|
| Quality of life | <--- | Obsessive- compulsive disorder | -.076 | .039 | -1.965 | .049 | -.068 |
| Quality of life | <--- | Health anxiety | -.338 | .106 | -3.177 | .001 | -.189 |
| Quality of life | <--- | Perceived stress | -1.035 | .071 | -14.642 | *** | -.980 |
| Environmental health | <--- | Quality of life | 1.000 | | | | .733 |
| Social relationships | <--- | Quality of life | .491 | .029 | 17.156 | *** | .742 |
| Psychological health | <--- | Quality of life | 1.073 | .051 | 20.867 | *** | .912 |
| Physical health | <--- | Quality of life | .941 | .054 | 17.319 | *** | .748 |
| Perceived helplessness | <--- | Perceived stress | 1.000 | | | | .648 |
| Perceived self-efficacy | <--- | Perceived stress | -.946 | .060 | -15.761 | *** | -.714 |
| Negative effect of disease | <--- | Health anxiety | 1.000 | | | | 1.007 |
| Risk of disease | <--- | Health anxiety | 1.564 | .436 | 3.588 | *** | .504 |
| Contamination obsessions | <--- | Obsessive-compulsive disorder | 1.423 | .678 | 2.099 | .036 | .859 |
| Washing compulsions | <--- | Obsessive- compulsive disorder | 1.000 | | | | .738 |

[34], Trougakos et al. [35], Korkmaz et al. [36], and Heidari Shams et al. [37]. Trougakos et al. [35] indicated that COVID-19 health anxiety impair health (somatic complaints), home (family engagement), and work (goal progress) outcomes due to increased emotion suppression and lack of psychological need fulfillment. In addition, individuals who frequently engage in handwashing behavior were buffered from the negative impact of COVID-19 health anxiety. Korkmaz et al. [36] found that the quality of life in health care workers who worked in the COVID-19 outpatient clinics or emergency departments, decreased due to increased levels of anxiety. Also, in Hayter et al.'s [34] study, the high health anxiety group reported poorer quality of life relative to the groups with relatively high or low health anxiety. In explaining this finding, it can be stated that the excessive increase in health anxiety in some people during the epidemic of viral diseases such as COVID-19 weakens the body's immune system against the disease and can affect the rational decisions and social behaviors of individuals. High levels of health anxiety are associated with catastrophizing the physical changes and the occurrence of maladaptive coping behaviors, leads to distress, social disability, occupational dysfunction, and frequent visits to healthcare centers [38], and as a result, people's quality of life will decrease.

## The correlation between perceived stress and quality of life

The results also revealed that perceived stress was significantly and negatively correlated with quality of life. This finding is consistent with previous studies conducted by Çelmeçe &

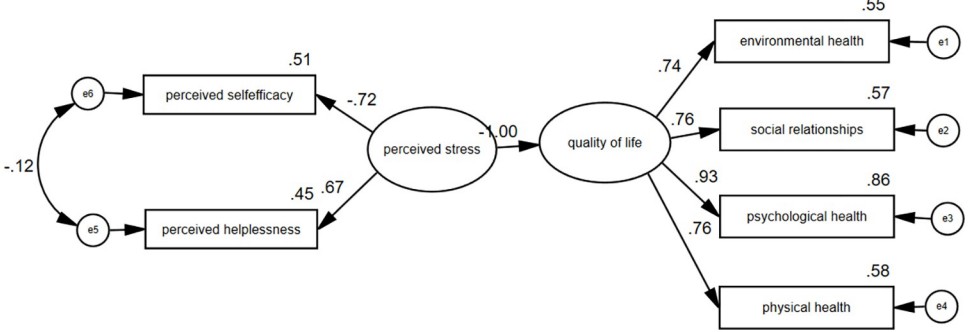

**Fig 2. The structural equation modeling (SEM) of the relationship between perceived stress and quality of life.**

**Table 6. Regression weight & standardized regression weight.**

|  |  |  | Estimate | S.E. | C.R. | P | Standardized Estimate |
|---|---|---|---|---|---|---|---|
| Quality of life | <--- | Perceived stress | -1.073 | .068 | -15.664 | *** | -1.000 |
| Environmental health | <--- | Quality of life | 1.000 |  |  |  | .744 |
| Social relationships | <--- | Quality of life | .494 | .027 | 17.979 | *** | .756 |
| Psychological health | <--- | Quality of life | 1.091 | .050 | 22.004 | *** | .927 |
| Physical health | <--- | Quality of life | .944 | .052 | 18.114 | *** | .762 |
| Perceived helplessness | <--- | Perceived stress | 1.000 |  |  |  | .668 |
| Perceived self-efficacy | <--- | Perceived stress | -.922 | .057 | -16.229 | *** | -.717 |

Menekay [39], Tejoyuwono et al. [40], and Qi et al. [12]. Studies suggest that perceived stress leads to higher levels of anxiety and lower levels of health-related quality of life [41]. The results of a study conducted by Çelmeçe & Menekay [39] on healthcare professionals caring for COVID-19 patients, showed that there was a high and negative correlation between stress and quality of life. Tejoyuwono et al. [40] found that the students' mental health status (anxiety, depression and stress) during the pandemic COVID-19 was dominantly on the normal category and the quality of life was moderate in all domains. Stress also significantly affected all domains of quality of life. In explaining this finding, it can be stated that prolonged quarantine, fear of disease, mental fatigue, insufficient information, financial loss, loss of freedom, uncertainty about the status of disease, uncertainty about time of disease control and the seriousness of the risk are among the stressors, which may affect the people's quality of life during this pandemic. On the other hand, when the situation is stressful and people fail to control the environmental stimuli, stress can negatively affect their well-being or performance.

## The correlation between obsessive-compulsive disorder and quality of life

The present results showed a significant negative correlation between obsessive-compulsive disorder and quality of life. In previous epidemics such as Acute Respiratory Syndrome (SARS), Middle East Respiratory Syndrome (MERS), and the influenza, exacerbations of OCD have been reported 6 to 12 months after its outbreak. When coping strategies involved repetitive behaviors, there is a risk of increasing obsessive-compulsive disorder [42]. Shaoo et al. [43] found that patients with OCD scored significantly less in total score as well as all the domains of quality of life compared to normal controls. The main characteristic of obsessive-compulsive disorder (OCD) is the presence of recurrent or severe obsessions or compulsions that cause considerable suffering for person. These obsessions cause significant disruption in the normal process of life, normal social activities, and relationships [44] and thus negatively affect a person's quality of life. OCD as a disorder affects the quality of life in domains of mental health such as social functioning and role limitation due to emotional problems and mental health [45].

## Study limitations

The first limitation of this study is that we selected participants to study through an online survey platform, those who did not have access to online surveys (e.g., the elderly), people who did not have access to the internet and people who were literate, entered the study less than others and the findings may be prone to selection bias. Second, psychological health was reported by the participants themselves, and no acceptable evaluation tool was used. Third, the data in this study were self-reported, and participants' responses may be prone to social desirability bias. Therefore, future studies are suggested to design and implement intervention

studies in the form of some models and theories such as social cognitive theory, protection motivation theory and stress management theory. Also, this study is recommended in other populations (such as patients recuperated from COVID-19 or individuals who have lost a family member due to COVID-19) and with other psychological variables (such as depression, Post traumatic stress disorder (PTSD) or prolonged grief disorder). Despite these possible limitations, the present study had some strong points, including data collection from 2 provinces of Iran (Bandar Abbas: a city with commercial ports and Isfahan: an industrial city and a city with a lot of tourist attractions, which these characteristics contribute to the high prevalence of COVID-19 in these cities). In addition, we used the validated questionnaires for data collection. These questionnaires were translated and validated in Persian by Iranian researchers. Finally, data collection coincided with pandemic peak time.

## Conclusion

The present findings showed that the outbreak of COVID-19 has changed people's living conditions and has had devastating psychological effects such as stress, health anxiety and obsessive behaviors, resulting in reduced quality of life. Thus, the design of psychological interventions to improve the mental health of individuals during the disease has a particular importance and the necessary measures should be taken by policy makers and relevant officials in this regard. It is necessary to implement counseling and psychological interventions at the community level in the current situation, interventions such as: creating support networks with the presence of a counselor and a psychologist, phone counseling, psychological support for medical staff, patients recuperated from Covid-19 or individuals who have lost a family member due to Covid-19 disease and so on.

## Supporting information

**S1 Data.**
(SAV)

## Acknowledgments

The authors would like to thank all the participants in the study.

## Author Contributions

**Conceptualization:** Atefeh Homayuni, Zahra Hosseini, Farah Moayedi.

**Data curation:** Atefeh Homayuni, Nahid Shahabi, Roghayeh Ezati Rad.

**Formal analysis:** Atefeh Homayuni.

**Investigation:** Atefeh Homayuni, Zahra Hosseini, Nahid Shahabi, Roghayeh Ezati Rad, Farah Moayedi.

**Methodology:** Atefeh Homayuni.

**Project administration:** Atefeh Homayuni.

**Supervision:** Zahra Hosseini.

**Validation:** Atefeh Homayuni, Zahra Hosseini.

**Writing – original draft:** Atefeh Homayuni, Nahid Shahabi.

**Writing – review & editing:** Atefeh Homayuni, Zahra Hosseini, Nahid Shahabi, Roghayeh Ezati Rad, Farah Moayedi.

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
