## [Decision Letter · Decision Letter 0]

11 Apr 2022

PONE-D-21-30356Quality of life and its related psychological problems during coronavirus pandemicPLOS ONE

Dear Dr. Hosseini,

Thank you for submitting your manuscript to PLOS ONE. After careful consideration, we feel that it has merit but does not fully meet PLOS ONE’s publication criteria as it currently stands. Therefore, we invite you to submit a revised version of the manuscript that addresses the points raised during the review process.

We look forward to receiving your revised manuscript.

Kind regards,

Mohammad Hossein Ebrahimi

Academic Editor

PLOS ONE

Journal Requirements:

3. Thank you for stating the following in the Acknowledgments Section of your manuscript: "The authors would like to acknowledge the financial support of the Hormozgan University of Medical Sciences. Also, we are grateful to the all participants."

Please remove any funding-related text from the manuscript and let us know how you would like to update your Funding Statement. Currently, your Funding Statement reads as follows: "The authors received no specific funding for this work."

6. Please ensure that you refer to Figure 1 in your text as, if accepted, production will need this reference to link the reader to the figure.

Reviewers' comments:

Reviewer's Responses to Questions

**Comments to the Author**

1. Is the manuscript technically sound, and do the data support the conclusions?

Reviewer #1: Yes

Reviewer #2: Partly

Reviewer #3: Yes

2. Has the statistical analysis been performed appropriately and rigorously? 

Reviewer #1: Yes

Reviewer #2: I Don't Know

Reviewer #3: Yes

3. Have the authors made all data underlying the findings in their manuscript fully available?

Reviewer #1: Yes

Reviewer #2: Yes

Reviewer #3: Yes

4. Is the manuscript presented in an intelligible fashion and written in standard English?

Reviewer #1: Yes

Reviewer #2: No

Reviewer #3: Yes

5. Review Comments to the Author

Reviewer #1: 1. Please highlight your contribution and novelty of this manuscript with accuracy in the introduction part before the arrangement description.

2. The literature and theoretical background and should improve and add more relevant studies e.g. (latest) to grab and display more contemporary literature critically.

3. Please check your manuscript for errors. On first glance at your manuscript, we noticed several grammar errors. Please check the whole paper carefully or have it professionally edited.

4. In the discussion part, you must contact the results, and some of the content needs to be rewritten.

5. It is suggested to use advanced statistical methods to add the influencing factors on the psychological changes of the population. For example, the potential influencing factors were analyzed by logitic regression

6. Note the order of Table 3 and table 4 in the article

Reviewer #2: This study entitled "Quality of life and its related psychological problems during coronavirus pandemic" aimed to assess that relationship in a population of adult people in Iran. This topic has been extensively covered in the literature over the past two years. The correlation between quality of life and mental health has been extensively studied to date. Although this study concerns a population very affected by the covid-19 pandemic, the sample studied does not seem to fully correspond to the population from which it is extracted. In addition, the demographic characteristics collected are not sufficient to give a complete view of the sample. A big lack of data concerns the period in which the study was conducted, or rather in which the responses from the participants were collected. In addition, the convenience sample and the method of recruiting participants represent possible selection biases (the recruitment method has not been clearly specified). Language must be deeply revised. The Discussion section lack of "linearity"; it is unclear, with statements followed by some bibliographical references, without a logic and concise semantic structure. The statystical results are also presentend in an unclear way. The declared limitations are quite slight, just as the few strengths of the study were not highlighted (for example the use of validated questionnaires). The conclusion, finally, lack of strenght and "interest".

Reviewer #3: Thank you for giving me the opportunity to review this article. Here are some questions to the authors.

I think this manuscript has a appropriate introduction and well justified.

Threre are some repetitive data and statements in the abstract (result) that is better to modify.

Participants: How authors are certain the sample population have no OCOD? I know it is mentioned in limitations but self disclosure is not enough.

I suggest that The importance of this study and its novelty should be explain more.

In my opinion statistical analyses are very complete and all well detailed.

In the discussion perhaps it is missing to indicate future lines of research.

---

## [Author Response · Author response to Decision Letter 0]

1 Jul 2022

Dear Editor in Chief, 

Journal of PLOS ONE

Thank you for giving us this opportunity to revise the manuscript entitled: "Quality of life and its related psychological problems during coronavirus pandemic". The corrections have been made in the manuscript based on Comments.

We used the yellow color to highlight the requested revisions by the honorable reviewer 1, the bright green to highlight the requested revisions by the honorable reviewer 2, the dark yellow color to highlight the requested revisions by the honorable reviewer 3, and the turquoise color to highlight the common requested revisions by all of the honorable reviewers. We also used the pink color to highlight the requested revisions (journal requirements).

I hope they would be satisfactory. 

I wish all the best for you

Shirin Hosseini, Ph.D

Corresponding Author

Reviewer Comments:

Honorable reviewer 1:

1.Please highlight your contribution and novelty of this manuscript with accuracy in the introduction part before the arrangement description.

• Our thanks to the honorable reviewer for your attention. Some explanations about the importance of this study and its novelty were presented in the last paragraph in the Introduction section. 

• Regarding the importance and novelty of this study and the necessity of doing it, the following can be mentioned:

1. Coronavirus disease 2019 is a new subject and its effects are diverse in different people. This study has examined various psychological variables and their relationships with quality of life during coronavirus pandemic (especially health anxiety and its relationship with obsessive-compulsive disorder) which has been less studied in other studies. This study also has examined stress and health anxiety separately and few studies have addressed these two types of emotional problems separately. 

2. This study concerns a population very affected by the Covid-19 pandemic. Vaccination against covid-19 started later in Iran than in other countries in the world. Delay in starting vaccination, people's negligence in observing health protocols, their resistance to vaccination and other reasons, led to higher prevalence and mortality due to the coronavirus in Iran, and as a result, imposed extra psychological pressure on people.

3. This research was conducted in two provinces of Hormozgan and Isfahan. These cities were more vulnerable to coronavirus, for some reasons: Isfahan is an industrial city and a city with a lot of tourist attractions and Hormozgan is a city with commercial ports. The hot weather, the impossibility of proper ventilation especially during the summer, and social gathering in closed environments led to the prevalence of covid-19 in Hormozgan. Besides, high humidity made it difficult for people to use face masks outdoors. In addition, in Iran, the cultural background of people, who are known to hospitable, was also influential in the increased prevalence of this disease. Generally, hospitality is more prominent in southern provinces of Iran, especially in Hormozgan province. All of this contributes to the high prevalence of covid-19 and further psychological problems in these cities.

2. The literature and theoretical background should improve and add more relevant studies e.g. (latest) to grab and display more contemporary literature critically.

• Our thanks to the honorable reviewer for your attention. According to the honorable reviewer's suggestion, the literature and theoretical background was improved and more relevant studies were added.

3. Please check your manuscript for errors. On first glance at your manuscript, we noticed several grammar errors. Please check the whole paper carefully or have it professionally edited.

• Our thanks to the honorable reviewer for your attention. According to the honorable reviewer's suggestion, the manuscript was professionally edited.

4. In the discussion part, you must contact the results, and some of the content needs to be rewritten.

• Our thanks to the honorable reviewer for your attention. Some changes made in the discussion section. We added sub-headings to the Discussion section to guide the readers through this section. We wrote the discussion section as follows: we summarized the important findings in the first paragraph. We presented the findings separately in several paragraphs and compared them with the results of previous studies. And finally we explained the obtained findings and explained why such a result was achieved.

5. It is suggested to use advanced statistical methods to add the influencing factors on the psychological changes of the population. For example, the potential influencing factors were analyzed by logistic regression.

• Our thanks to the honorable reviewer for your attention. We used path analysis and structural equation modeling (SEM) to test the relationships between the variables. Regarding the suggestion of honorable reviewer (using logistic regression), it should be noted that logistic regression is used to predict a multi-category dependent variable. However, "quality of life" as a dependent variable in this study, is not a multi-category variable. Based on the explanations provided, does the honorable reviewer suggest a specific statistical test? 

6. Note the order of Table 3 and table 4 in the article.

• Our thanks to the honorable reviewer for your attention. The order of table 3 and table 4 in the article was modified.

Honorable Reviewer 2:

1.This study entitled "Quality of life and its related psychological problems during coronavirus pandemic" aimed to assess that relationship in a population of adult people in Iran. This topic has been extensively covered in the literature over the past two years. The correlation between quality of life and mental health has been extensively studied to date. Although this study concerns a population very affected by the covid-19 pandemic, the sample studied does not seem to fully correspond to the population from which it is extracted. 

• Our thanks to the honorable reviewer for your attention. Regarding the importance and novelty of this study and the necessity of doing it, the following can be mentioned:

4. Coronavirus disease 2019 is a new subject and its effects are diverse in different people. This study has examined various psychological variables and their relationships with quality of life during coronavirus pandemic (especially health anxiety and its relationship with obsessive-compulsive disorder) which has been less studied in other studies. This study also has examined stress and health anxiety separately and few studies have addressed these two types of emotional problems separately. 

5. As the honorable reviewer pointed out, this study concerns a population very affected by the Covid-19 pandemic. Vaccination against covid-19 started later in Iran than in other countries in the world. Delay in starting vaccination, people's negligence in observing health protocols, their resistance to vaccination, economic problems caused by the coronavirus and other reasons, led to higher prevalence and mortality due to the coronavirus in Iran, and as a result, imposed extra psychological pressure on people.

6. This research was conducted in two provinces of Hormozgan and Isfahan. These cities were more vulnerable to coronavirus, for some reasons: Isfahan is an industrial city and a city with a lot of tourist attractions and Hormozgan is a city with commercial ports. The hot weather, the impossibility of proper ventilation especially during the summer, and social gathering in closed environments led to the prevalence of covid-19 in Hormozgan. Besides, high humidity made it difficult for people to use face masks outdoors. In addition, in Iran, the cultural background of people, who are known to hospitable, was also influential in the increased prevalence of this disease. Generally, hospitality is more prominent in southern provinces of Iran, especially in Hormozgan province. All of this contributes to the high prevalence of covid-19 and further psychological problems in these cities.

Some explanations about the importance of this study and its novelty were presented in the last paragraph in the Introduction section.

2. In addition, the demographic characteristics collected are not sufficient to give a complete view of the sample. 

• Our thanks to the honorable reviewer for your attention. The most important demographic characteristics related to the purpose of our study (and based on similar studies) were collected. Does the honorable reviewer have a special demographic characteristic in mind?

3.A big lack of data concerns the period in which the study was conducted, or rather in which the responses from the participants were collected.

• Our thanks to the honorable reviewer for your attention. The data collection was conducted from October 22, 2020 to February 3, 2021.

4.In addition, the convenience sample and the method of recruiting participants represent possible selection biases (the recruitment method has not been clearly specified).

• Our thanks to the honorable reviewer for your attention. Further details on data collection are provided in Materials and Methods (Sample size and sampling procedure section). It should be noted that due to the prevalence of coronavirus and it severity during the study (data collection) in Iran and the need to compliance with quarantine, we had to collect data through social media. Otherwise, both the researchers and the participants would have to pay heavy fines. In study limitations, we have mentioned that one of the limitations of this study is that the questionnaire survey was conducted only via the internet, which was only available to participants who were literate and had access to the internet.

5.Language must be deeply revised. 

• Our thanks to the honorable reviewer for your attention. According to the honorable reviewer's suggestion, language was deeply revised.

6.The Discussion section lack of "linearity"; it is unclear, with statements followed by some bibliographical references, without a logic and concise semantic structure. 

• Our thanks to the honorable reviewer for your attention. Some changes made in the discussion section. We added sub-headings to the Discussion section to guide the readers through this section. We wrote the discussion section as follows: we summarized the important findings in the first paragraph. We presented the findings separately in several paragraphs and compared them with the results of previous studies. And finally we explained the obtained findings and explained why such a result was achieved.

7. The statistical results are also presented in an unclear way. 

• Our thanks to the honorable reviewer for your attention. Some explanations about the results presented in each table, are provided at the bottom of the table.

8. The declared limitations are quite slight, just as the few strengths of the study were not highlighted (for example the use of validated questionnaires). 

• Our thanks to the honorable reviewer for your attention. We highlighted the study strengths and modified the study limitations.

9. The conclusion, finally, lack of strength and "interest".

• Our thanks to the honorable reviewer for your attention. We made some changes in the conclusion section.

Honorable Reviewer 3:

Thank you for giving me the opportunity to review this article. Here are some questions to the authors. I think this manuscript has an appropriate introduction and well justified. 

1.There are some repetitive data and statements in the abstract (result) that is better to modify.

Our thanks to the honorable reviewer for your attention. According to the honorable reviewer's suggestion, the repetitive data and statements in the Abstract (result section) was modified.

2. Participants: How authors are certain the sample population have no OCOD? I know it is mentioned in limitations but self-disclosure is not enough.

• Our thanks to the honorable reviewer for your attention. It should be noted that individuals who had no history of taking any psychiatric medication were included in the study. In other words, the authors examined this criterion (not suffering from psychological disorders such as obsessive-compulsive disorder) by asking this question: "Do you have a history of taking psychiatric medications?". 

3. I suggest that The importance of this study and its novelty should be explain more.

• Our thanks to the honorable reviewer for your attention. Some explanations about the importance of this study and its novelty were presented in the last paragraph in the Introduction section.

• Regarding the importance and novelty of this study and the necessity of doing it, the following can be mentioned:

1.Coronavirus disease 2019 is a new subject and its effects are diverse in different people. This study has examined various psychological variables and their relationships with quality of life during coronavirus pandemic (especially health anxiety and its relationship with obsessive-compulsive disorder) which has been less studied in other studies. This study also has examined stress and health anxiety separately and few studies have addressed these two types of emotional problems separately. 

2.This study concerns a population very affected by the Covid-19 pandemic. Vaccination against covid-19 started later in Iran than in other countries in the world. Delay in starting vaccination, people's negligence in observing health protocols, their resistance to vaccination and other reasons, led to higher prevalence and mortality due to the coronavirus in Iran, and as a result, imposed extra psychological pressure on people.

3.This research was conducted in two provinces of Hormozgan and Isfahan. These cities were more vulnerable to coronavirus, for some reasons: Isfahan is an industrial city and a city with a lot of tourist attractions and Hormozgan is a city with commercial ports. The hot weather, the impossibility of proper ventilation especially during the summer, and social gathering in closed environments led to the prevalence of covid-19 in Hormozgan. Besides, high humidity made it difficult for people to use face masks outdoors. In addition, in Iran, the cultural background of people, who are known to hospitable, was also influential in the increased prevalence of this disease. Generally, hospitality is more prominent in southern provinces of Iran, especially in Hormozgan province. All of this contributes to the high prevalence of covid-19 and further psychological problems in these cities.

4. In my opinion statistical analyses are very complete and all well detailed.

• Our thanks to the honorable reviewer for your attention.

5. In the discussion perhaps it is missing to indicate future lines of research.

• Our thanks to the honorable reviewer for your attention. Some suggestions for future studies were presented in the discussion section, at the end of "study limitations" part.

---

## [Decision Letter · Decision Letter 1]

4 Oct 2022

PONE-D-21-30356R1Quality of life and its related psychological problems during coronavirus pandemicPLOS ONE

Dear Dr. Hosseini,

Thank you for submitting your manuscript to PLOS ONE. After careful consideration, we feel that it has merit but does not fully meet PLOS ONE’s publication criteria as it currently stands. Therefore, we invite you to submit a revised version of the manuscript that addresses the points raised during the review process.

We look forward to receiving your revised manuscript.

Kind regards,

Mohammad Hossein Ebrahimi

Academic Editor

PLOS ONE

Reviewers' comments:

Reviewer's Responses to Questions

**Comments to the Author**

1. If the authors have adequately addressed your comments raised in a previous round of review and you feel that this manuscript is now acceptable for publication, you may indicate that here to bypass the “Comments to the Author” section, enter your conflict of interest statement in the “Confidential to Editor” section, and submit your "Accept" recommendation.

Reviewer #4: (No Response)

Reviewer #5: (No Response)

Reviewer #6: (No Response)

2. Is the manuscript technically sound, and do the data support the conclusions?

Reviewer #4: Partly

Reviewer #5: Partly

Reviewer #6: Partly

3. Has the statistical analysis been performed appropriately and rigorously? 

Reviewer #4: No

Reviewer #5: Yes

Reviewer #6: Yes

4. Have the authors made all data underlying the findings in their manuscript fully available?

Reviewer #4: Yes

Reviewer #5: No

Reviewer #6: No

5. Is the manuscript presented in an intelligible fashion and written in standard English?

Reviewer #4: No

Reviewer #5: No

Reviewer #6: Yes

6. Review Comments to the Author

Reviewer #4: Dear Authors,

I am pleased to study and review your study. The English proficiency of the manuscript is not appropriate, it should be re-written. I addressed many of them for your consideration. The statistical analysis is conducted incorrectly, you need to re-run the analyses as suggested and then write the results and discussion from scratch.

Good luck with the revision.

Reviewer #5: The study is an epidemiological cross-sectional exploratory study, looking into COVID-19-related anxiety, obsessions, and their link with the general population’s perceived quality of life. The sample size and characteristics make the results adequately generalizable. Although the study design, procedures, and findings do not show any novel features, the research question (if stated clearly) is well worth investigating among different populations and nationalities.

There are some minor concerns and comments which could improve the manuscript:

• Keywords are rather irrelevant to the main aims of the study and are either too general or too specific.

• Some typos across the manuscript – for instance, line 148, and 405.

• The introduction is sufficiently elaborative. However, the aims and hypotheses of this study are missing. I understand that it was a rather exploratory study, but adding the pre-assumptions and research questions could add a lot of value to your introduction. Why is this study conducted?

• A self-report measure might not be the best way to exclude the participants with psychological disorders (e.g., OCD) ¬– it would have been better if you used a screening tool to exclude participants with psychiatric symptoms of OCD.

• You used a snowball sampling method (as stated in line 177) – could you also report what standards you had for the snowballing? Were the participants aware of the inclusion/exclusion criteria for choosing whom to share the questionnaire with?

• What do you mean by the ‘underlying diseases’ in line 195, and how did you measure them?

• Table 3 is an unnecessarily exhaustive way of showing the correlation coefficients. You could perhaps summarize this table or convert it to a figure – namely a heat map.

• Figure 1 has a low resolution and quality. Could it be improved resolution and graphical-wise? It could also benefit from a more informative caption.

• I recommend separating the results section into subsections making it easier to follow. (could be classified as demographics, correlations, standardized estimations, etc.).

• There are some inconsistencies across the manuscript regarding the name of the virus (coronavirus, COVID, Covid-19) – try to use a consistent term across the entire text.

• I recommend reformatting the study strengths section to a text form rather than bullet-pointed items with possibly more elaborations (like why collecting data from a city with tourist attractions is considered a strength of this study?).

Overall, the study is an interesting one in light of the global complexities resulting from the pandemic and its influences on people's well-being. The manuscript could also benefit from a native English speaker for proofreading and revising some parts language-wise.

Reviewer #6: The research itself is well designed to test the interrelationships among the different variables of psychological distress during the Covid 19 pandemic in the Iranian population. The data collection tools and studied population were selected appropriately. However, a few points in the paper require further elaboration. This paper can be published after incorporating a few elements outlined below:

In Abstract:

Results: please provide the value of r in decimals and mention the exact p-value. For example: (r = - or + 0. xx, p = 0.xxx) rather than just saying significant positive or negative correlations. The exact p-value is mandatory (look at the APA manual on reporting of r values). For reference, you can find similar information in this paper - https://www.sciencedirect.com/science/article/pii/S2452247318302164

In the main text:

1. The use of the language of the tools (short health anxiety inventory (SHAI), perceived stress scale (PSS), world health 52 organization quality of life questionnaire (WHOQOL-BREF) and Padua inventory)

- What was the language used for data collection? Persian/Farsi or English?

In the strength of the research, the authors have claimed that they have used validated tools. Were those tools validated in English or the local language? I believe that not all citizens of Iran could communicate well in the English language, if this is the case, the use of tools in the local language needs to have proper validation of the tools. The validation of tools in the local language is a long procedure and demands rigorous processes. In many cases, the authors have mentioned several studies have explored this and that, but no concrete findings or studies with explicit information on the language used, the validation process was done etc. were covered.

2. The collection of data from minors (below the age of 18) without the consent of their parents is a topic researcher should be aware of and avoid in any behavioural research. The information on the total number of children surveyed is not disclosed. It would be nice to clarify this topic.

3. Study strengths: I would recommend describing them in narratives rather than just putting bullet points. Kindly correct the validated tools-related information there if it was not the case.

4. I was concerned on the exclusion criteria also included people with psychological disorders such as obsessive-compulsive disorder and unwillingness to participate in the study. Please clarify what was the process of identifying these disorders. was it the beneficiary’s self-disclosure or did the researchers assess their disorders before the research participation? The process is not mentioned in the description. However, it would be nice to avoid such situations as disclosing personal mental health status just with researchers without a follow-up on the situation of the respondent is often not a fair treatment to the research participants. It is recommended to clarify this issue, whether there was any debriefing or message with resources on seeking mental health services or not.

In summary, the research is well designed, the data analysis process and findings of the research are well articulated, and the research findings can benefit other researchers/service providers in future. Thanks to the efforts of the authors for their efforts in this research.

---

## [Author Response · Author response to Decision Letter 1]

13 Oct 2022

Dear Editor in Chief, 

Journal of PLOS ONE

Thank you for giving us this opportunity to revise the manuscript entitled: "Quality of life and its related psychological problems during coronavirus pandemic". The corrections have been made in the manuscript based on Comments.

We used the yellow color to highlight the requested revisions by the honorable reviewer 4, the bright green to highlight the requested revisions by the honorable reviewer 5, the pink color to highlight the requested revisions by the honorable reviewer 6, and the turquoise color to highlight the common requested revisions by all of the honorable reviewers. 

It should be noted that in the first revision, the honorable reviewer 2 asked us to add the strengths of the study to the conclusion. According to the honorable reviewer's suggestion, we added the "study strengths". But in the latest revision, the honorable reviewer 4 commented that "there is no study strengths section in scientific papers and this section should be removed". Meanwhile, two other honorable reviewers have emphasized the study strengths. According to these honorable reviewers' suggestion, we added the "study strengths" following the "study limitations" in the recent revision. We request help from the honorable editor regarding the removal or addition of the "study strengths."

I hope they would be satisfactory. 

I wish all the best for you

Shirin Hosseini, Ph.D

Corresponding Author

Reviewer #4: Dear Authors,

I am pleased to study and review your study. The English proficiency of the manuscript is not appropriate, it should be re-written. I addressed many of them for your consideration. The statistical analysis is conducted incorrectly, you need to re-run the analyses as suggested and then write the results and discussion from scratch.

Good luck with the revision.

• Our thanks to the honorable reviewer for your complimentary comments and suggestions. According to the honorable reviewer's suggestions, all of the suggestions were applied in the manuscript and highlighted with yellow color. Language was deeply revised and the statistical analysis was run again as suggested. 

• Regarding this comment of the honorable reviewer "You clearly mentioned that you have measured the correlation between quality of life and obsessive-compulsive disorder, then it should be an inclusion criteria!" at page 17, it should be noted that our purpose of considering this inclusion criterion was to exclude people from the study who were suffering from psychological disorders (such as OCD) even before the spread of Coronavirus. We wanted to investigate the prevalence of psychological disorders after the spread of Coronavirus and evaluate its impact on people's quality of life. So one of the inclusion criteria in our study was "not suffering from psychological disorders such as obsessive-compulsive disorder". We assess this criterion by a self-report question at the beginning of the questionnaire. Since we did not have access to the medical records of the participants, we relied on their self-reports (people who suffer from psychological disorders are mostly aware of their illness, and we assumed that these people are honest in answering this question regarding having psychological disorders). Individuals who confirmed that they do not have psychological disorders, their answers to the questions were examined and analyzed. 

• Regarding this comment of the honorable reviewer "These could not be considered as study strengths. The whole section should be removed, there is no "study strengths" section in scientific papers. If you want to emphasis on the importance of your study you need to add it to introduction section.", it should be noted that in the first revision, the honorable reviewer 2 asked us to add the strengths of the study to the conclusion and in this revision, two other honorable reviewers have emphasized the study strengths. According to these honorable reviewers' suggestion, we added the "study strengths" following the "study limitations" in the recent revision. According to what has been said, what is the order of the honorable reviewer in this regard?

Reviewer #5: The study is an epidemiological cross-sectional exploratory study, looking into COVID-19-related anxiety, obsessions, and their link with the general population’s perceived quality of life. The sample size and characteristics make the results adequately generalizable. Although the study design, procedures, and findings do not show any novel features, the research question (if stated clearly) is well worth investigating among different populations and nationalities.

There are some minor concerns and comments which could improve the manuscript:

1-Keywords are rather irrelevant to the main aims of the study and are either too general or too specific.

• Our thanks to the honorable reviewer for your attention. All keywords were set based on MeSH. Does the honorable reviewer suggest special keywords?

2- Some typos across the manuscript – for instance, line 148, and 405.

• Our thanks to the honorable reviewer for your attention. These typos were corrected.

3- The introduction is sufficiently elaborative. However, the aims and hypotheses of this study are missing. I understand that it was a rather exploratory study, but adding the pre-assumptions and research questions could add a lot of value to your introduction. Why is this study conducted?

• Our thanks to the honorable reviewer for your attention. According to the honorable reviewer's suggestion, the specific research question and hypotheses of this study were written at the end of the Introduction section.

4- A self-report measure might not be the best way to exclude the participants with psychological disorders (e.g., OCD) ¬– it would have been better if you used a screening tool to exclude participants with psychiatric symptoms of OCD.

• Our thanks to the honorable reviewer for your attention. Since we did not have access to the medical records of the participants, we relied on their self-reports (people who suffer from psychological disorders are mostly aware of their illness, and we assumed that these people are honest in answering this question regarding having psychological disorders). We also used Padua inventory to assess OCD.

5-You used a snowball sampling method (as stated in line 177) – could you also report what standards you had for the snowballing? Were the participants aware of the inclusion/exclusion criteria for choosing whom to share the questionnaire with?

• Our thanks to the honorable reviewer for your attention. It should be noted that we used the cluster sampling method to collect data. This sentence was translated incorrectly. The residents of Isfahan participating in the study were 2 times more than the participants of Bandar Abbas.

6- What do you mean by the ‘underlying diseases’ in line 195, and how did you measure them?

• Our thanks to the honorable reviewer for your attention. The "underlying diseases" mean "chronic diseases" such as hypertension, diabetes, cardiovascular disease, kidney disease and etc. Because people who are suffering from chronic diseases are more exposed to COVID-19 and its complications than healthy people. We measured this item by a self-report question.

7- Table 3 is an unnecessarily exhaustive way of showing the correlation coefficients. You could perhaps summarize this table or convert it to a figure – namely a heat map.

• Our thanks to the honorable reviewer for your attention. We calculated the correlation coefficients only for main variables, and subscales of quality of life were removed. 

8- Figure 1 has a low resolution and quality. Could it be improved resolution and graphical-wise? It could also benefit from a more informative caption.

• Our thanks to the honorable reviewer for your attention. We changed the caption of the figure. Does the honorable reviewer suggest a special and better caption? 

9- I recommend separating the results section into subsections making it easier to follow. (could be classified as demographics, correlations, standardized estimations, etc.).

• Our thanks to the honorable reviewer for your attention. According to the honorable reviewer's suggestion, we separated the results section into subsections.

10- There are some inconsistencies across the manuscript regarding the name of the virus (coronavirus, COVID, Covid-19) – try to use a consistent term across the entire text.

• Our thanks to the honorable reviewer for your attention. According to the honorable reviewer's suggestion, we used a consistent term (COVID-19) for the name of the virus across the entire text.

11- I recommend reformatting the study strengths section to a text form rather than bullet-pointed items with possibly more elaborations (like why collecting data from a city with tourist attractions is considered a strength of this study?).

• Our thanks to the honorable reviewer for your attention. According to the honorable reviewer's suggestion, we reformatted the study strengths section to a text form with more elaborations. This section is followed by the "Study limitations" section. Regarding this study strength "collecting data from 2 provinces of Iran", it should be noted that these cities were more vulnerable to coronavirus, for some reasons: Isfahan is an industrial city and a city with a lot of tourism attractions and Bandar Abbas is a city with commercial ports. Due to its commercial ports, Banda Abbas has many immigrants from other provinces. The hot weather, the impossibility of proper ventilation especially during the summer, and social gathering in closed environments led to the prevalence of covid-19 in Bandar Abbas. Besides, high humidity made it difficult for people to use face masks outdoors. In addition, in Iran, the cultural background of people (who are known to hospitable), placing a lot of tourist attractions in Isfahan and the beautiful beaches in Bandar Abbas, has attracted more tourists and passengers to these cities, which is influential in the increased prevalence of this disease. All of this contributes to the high prevalence of covid-19 and further psychological problems in these cities. (Due to the prolongation of "limitations" section, we have presented these elaborations there. At the discretion of the honorable reviewer, these elaborations will be added to the manuscript.)

Overall, the study is an interesting one in light of the global complexities resulting from the pandemic and its influences on people's well-being. The manuscript could also benefit from a native English speaker for proofreading and revising some parts language-wise.

• Our thanks to the honorable reviewer for your complimentary comments and suggestions. According to the honorable reviewer's suggestion, language was deeply revised.

Reviewer #6: The research itself is well designed to test the interrelationships among the different variables of psychological distress during the Covid 19 pandemic in the Iranian population. The data collection tools and studied population were selected appropriately. However, a few points in the paper require further elaboration. This paper can be published after incorporating a few elements outlined below:

In Abstract: Results: please provide the value of r in decimals and mention the exact p-value. For example: (r = - or + 0. xx, p = 0.xxx) rather than just saying significant positive or negative correlations. The exact p-value is mandatory (look at the APA manual on reporting of r values). For reference, you can find similar information in this paper - https://www.sciencedirect.com/science/article/pii/S2452247318302164

• Our thanks to the honorable reviewer for your attention. According to the honorable reviewer's suggestion, we mentioned the exact p-value.

In the main text:1. The use of the language of the tools (short health anxiety inventory (SHAI), perceived stress scale (PSS), world health 52 organization quality of life questionnaire (WHOQOL-BREF) and Padua inventory)

- What was the language used for data collection? Persian/Farsi or English?

In the strength of the research, the authors have claimed that they have used validated tools. Were those tools validated in English or the local language? I believe that not all citizens of Iran could communicate well in the English language, if this is the case, the use of tools in the local language needs to have proper validation of the tools. The validation of tools in the local language is a long procedure and demands rigorous processes. In many cases, the authors have mentioned several studies have explored this and that, but no concrete findings or studies with explicit information on the language used, the validation process was done etc. were covered.

• Our thanks to the honorable reviewer for your attention. We used Persian language for data collection and all of the tools were validated in Persian (Shams et al, 2010; Karimi et al, 2015; Usefy et al, 2010; Maroufizadeh et al, 2014). More explanations about each questionnaire were presented in the "Research tools" section.

2. The collection of data from minors (below the age of 18) without the consent of their parents is a topic researcher should be aware of and avoid in any behavioural research. The information on the total number of children surveyed is not disclosed. It would be nice to clarify this topic.

• Our thanks to the honorable reviewer for your attention. We obtained consent from the participants themselves. Because in Iran, the population aged 15 and over is considered as an adult population. Therefore, according to the instructions of the Ethics Committee of the Hormozgan University of Medical Sciences, for participants aged 16-18, this consent is received from themselves and not from their parents. In other words, for participants between 16-18 years of old, the research ethics committee waived the requirement for parental consent. If we understood the meaning of the honorable reviewer correctly, 24 participants were below the age of 18.

3. Study strengths: I would recommend describing them in narratives rather than just putting bullet points. Kindly correct the validated tools-related information there if it was not the case.

• Our thanks to the honorable reviewer for your attention. We reformatted the study strengths section to a text form with more elaborations. This section is followed by the "Study limitations" section. It should be noted that the tools used in this study were translated and validated in Persian by Iranian researchers ("Research tools" section).

4. I was concerned on the exclusion criteria also included people with psychological disorders such as obsessive-compulsive disorder and unwillingness to participate in the study. Please clarify what was the process of identifying these disorders. was it the beneficiary’s self-disclosure or did the researchers assess their disorders before the research participation? The process is not mentioned in the description. However, it would be nice to avoid such situations as disclosing personal mental health status just with researchers without a follow-up on the situation of the respondent is often not a fair treatment to the research participants. It is recommended to clarify this issue, whether there was any debriefing or message with resources on seeking mental health services or not.

• Our thanks to the honorable reviewer for your attention. We assess this criterion by a self-report question at the beginning of the questionnaire. Since we did not have access to the medical records of the participants, we relied on their self-reports (people who suffer from psychological disorders are mostly aware of their illness, and we assumed that these people are honest in answering this question regarding having psychological disorders). Individuals who confirmed that they do not have psychological disorders, their answers to the questions were examined and analyzed. If we understood the meaning of the honorable reviewer correctly regarding the last part of this comment, it should be noted that at the end of the questionnaire, in the "thanksgiving" section, we asked the participants (who were willing to know the result of their test) to enter their contact number in the box. After evaluating their responses, they were informed of their test's results and received a free phone consultation. 

In summary, the research is well designed, the data analysis process and findings of the research are well articulated, and the research findings can benefit other researchers/service providers in future. Thanks to the efforts of the authors for their efforts in this research.

• Our thanks to the honorable reviewer for your complimentary comments and suggestions.

---

## [Editor Report · Decision Letter 2]

17 Oct 2022

Quality of life and its related psychological problems during coronavirus pandemic

PONE-D-21-30356R2

Dear Dr. Hosseini,

We’re pleased to inform you that your manuscript has been judged scientifically suitable for publication and will be formally accepted for publication once it meets all outstanding technical requirements.

Kind regards,

Mohammad Hossein Ebrahimi

Academic Editor

PLOS ONE
---

## [Editor Report · Acceptance letter]

20 Oct 2022

PONE-D-21-30356R2 

Quality of life and its related psychological problems during coronavirus pandemic 

Dear Dr. Hosseini:

I'm pleased to inform you that your manuscript has been deemed suitable for publication in PLOS ONE. Congratulations! Your manuscript is now with our production department. 

Kind regards, 

on behalf of

Dr. Mohammad Hossein Ebrahimi 

Academic Editor

PLOS ONE